# Layer Analysis Based on RNA-Seq Reveals Molecular Complexity of Gastric Cancer

**DOI:** 10.3390/ijms252111371

**Published:** 2024-10-22

**Authors:** Pablo Perez-Wert, Sara Fernandez-Hernandez, Angelo Gamez-Pozo, Marina Arranz-Alvarez, Ismael Ghanem, Rocío López-Vacas, Mariana Díaz-Almirón, Carmen Méndez, Juan Ángel Fresno Vara, Jaime Feliu, Lucia Trilla-Fuertes, Ana Custodio

**Affiliations:** 1Department of Medical Oncology, Hospital Universitario La Paz, Paseo de la Castellana 261, 28046 Madrid, Spain; pablopwert@gmail.com (P.P.-W.); isma_g_c@hotmail.com (I.G.); jaimefeliu@hotmail.com (J.F.); 2Molecular Oncology Laboratory, Institute of Medical and Molecular Genetics-INGEMM, Hospital Universitario La Paz-IdiPAZ, Paseo de la Castellana 261, 28046 Madrid, Spain; sarafdezhernandez@gmail.com (S.F.-H.); angelogamez@gmail.com (A.G.-P.); rlvacas@salud.madrid.org (R.L.-V.); jafresno@gmail.com (J.Á.F.V.); 3IdiPAZ Biobank, La Paz University Hospital-IdiPAZ, Paseo de la Castellana 261, 28046 Madrid, Spain; biobanco.hulp@salud.madrid.org; 4Biostatistics Unit, La Paz University Hospital-IdiPAZ, Paseo de la Castellana 261, 28046 Madrid, Spain; mariana.diaz@salud.madrid.org; 5Department of Pathology, Hospital Universitario La Paz, 28046 Madrid, Spain; mcmendez1970@gmail.com; 6Biomedical Research Networking Center on Oncology-CIBERONC, ISCIII (Instituto de Salud Carlos III), 28029 Madrid, Spain; 7Cátedra UAM-AMGEN, Universidad Autónoma de Madrid, 28046 Madrid, Spain; 8Medicine Department, Universidad Autónoma de Madrid, 28046 Madrid, Spain

**Keywords:** gastric adenocarcinoma, TCGA subtypes, molecular classification, consensus cluster analysis, CIN-MSI-like group

## Abstract

Gastric adenocarcinoma (GA) is a significant global health issue with poor prognosis, despite advancements in treatment. Although molecular classifications, such as The Cancer Genome Atlas (TCGA), provide valuable insights, their clinical utility remains limited. We performed a multi-layered functional analysis using TCGA RNA sequencing data to better define molecular subtypes and explore therapeutic implications. We reanalyzed TCGA RNA-seq data from 142 GA patients with localized disease who received adjuvant chemotherapy. Our approach included probabilistic graphical models and recurrent sparse k-means/consensus cluster algorithms for layer-based analysis. Our findings revealed survival differences among TCGA groups, with the GS subtype showing the poorest prognosis. We identified twelve functional nodes and seven biological layers, each with distinct functions. The combined molecular layer (CML) classification identified three prognostic groups that align with TCGA subtypes. CML2 (GS-like) displayed gene expression related to lipid metabolism, correlating with worse survival. Transcriptomic heterogeneity within the CIN subtype revealed clusters tied to proteolysis and lipid metabolism. We identified a subset of CIN tumors with profiles similar to MSI, termed CIN-MSI-like. Claudin-18, a key gene in proteolysis, was overexpressed across TCGA subtypes, suggesting it is a potential therapeutic target. Our study advances GA biology, enabling refined stratification and personalized treatment. Further studies are needed to translate these findings into clinical practice.

## 1. Introduction

Gastric adenocarcinoma (GA) is a significant global health concern, with over 1 million estimated new cases and 768,793 cancer-related deaths in 2020 worldwide [1]. Most patients are diagnosed at advanced stages, resulting in high mortality rates. Although surgery is the cornerstone of treatment with curative intent for resectable locally advanced disease, defined as clinical stage ≥T2 and/or positive regional lymph node metastasis by AJCC-UICC staging system (8th Edition) [2], the 5-year survival rate remains poor [3]. As a result, various multimodal perioperative approaches, including perioperative chemotherapy (CT) [4], adjuvant CT [5], and chemoradiotherapy (CRT) [6], have been explored, leading to better long-term prognosis compared to mere surgical resection. Recently, the FLOT perioperative regimen has been postulated as the standard approach for locally advanced GA [7]. For gastroesophageal junction (GEJ) tumors, neoadjuvant CRT may also be an effective option [8]. Despite these advancements, therapeutic resistance continues to be a major obstacle, and cure rates remain low, with only approximately 35% of patients surviving at 5 years in the case of lymph node involvement. Further research and personalized approaches are therefore imperative to improve patient outcomes.

Several clinicopathological factors have been studied to identify patients who may benefit from chemotherapy, including immune ratios, systemic inflammation, tumor characteristics, and lymph node involvement. Promising molecular markers include MSI, inflammatory infiltrates, immune checkpoints, and ctDNA [9,10]. However, few are routinely used in clinical practice.

From a molecular standpoint, GA presents a high degree of molecular heterogeneity with subtypes exhibiting diverse biological behaviors. The Cancer Genome Atlas (TCGA) [11] proposes a molecular classification dividing GA into four subtypes with distinct clinical and prognostic features based on genetic and epigenetic data across six platforms: (1) Epstein–Barr virus (EBV)-positive tumors (9% of cases), which show recurrent PIK3CA mutations, DNA hypermethylation, and JAK2 or PD-L2 amplification; (2) Microsatellite unstable tumors (22% of cases), which display elevated mutation rates; (3) genomically stable (GS) tumors (20% of cases), which are enriched for the diffuse histological variant and RHOA mutations or fusions involving RHO-family GTPase-activating proteins; and (4) tumors with chromosomal instability (CIN) (50% of cases), which show marked aneuploidy and focal amplification of receptor tyrosine kinases. The Asian Cancer Research Group (ACRG) [12] established an alternative classification based on data solely from Asian patients, who were categorized into four molecular subtypes: (1) mesenchymal-like type, which includes diffuse-subtype tumors with the worst prognosis; (2) Microsatellite unstable tumors, which are hypermutated intestinal-subtype tumors with the best overall prognosis; and (3) TP53-active and (4) TP53-inactive types, which include patients with intermediate prognosis.

Several studies [12,13,14,15,16] have investigated the clinical characteristics and prognostic role of these molecular subgroups in different GA populations. They conclude that the GS subtype correlates quite well with the diffuse histology and has the poorest prognosis. The MSI subtype, more common in older patients, has the best prognosis along with the EBV subtype, which tends to be located proximally and is associated with intense inflammatory infiltration. The CIN subtype would be the most common and has an intermediate prognosis. Furthermore, some studies assessing the effectiveness of treatments according to molecular subtypes have concluded that the MSI and EBV groups are likely the ones that benefit the least from adjuvant CT [16,17] but are predictive factors for immunotherapy response. However, despite the growing body of knowledge, treatment decision-making is not currently based on molecular subtypes.

Recent advances in RNA sequencing make it possible to characterize transcriptionally diverse GA subgroups, which allows for better carcinogenesis understanding, more precise diagnosis, and the assessment of new prognostic and therapeutic strategies [13,14,18,19]. RNA-seq plays a pivotal role in incorporating omics data and is one of the platforms used to define molecular subgroups of GA within the TCGA classification [11]. In addition, computational analyses on intricate omics data enable a more profound comprehension of tumor molecular characteristics.

We conducted this study to obtain a deeper molecular classification of GA using RNA sequencing data from TCGA in patients with localized tumors who received adjuvant CT. Our aim is to gain a better understanding of the molecular heterogeneity of GA, thus allowing a patient’s classification in a way that provides improved clinical and therapeutic utility.

## 2. Results

### 2.1. Gastric Adenocarcinoma TCGA Cohort

The GA TCGA cohort includes 443 patients with clinical data available. A homogeneous group of 150 patients with localized disease who were treated with 5-fluorouracil and/or platinum-based adjuvant CT was selected for further analyses. Eight of these patients with no RNA-seq data were excluded. Therefore, the final study population included 142 patients, 88 of whom had the TCGA molecular subtype available and 54 patients without this information. The patient selection process is illustrated in Appendix A.

At a median follow-up of 15 months, median overall survival (OS) has not been reached with 25 deaths occurring. Our analysis revealed significant distinct survival outcomes among TCGA groups, with the GS subtype showing the worst prognosis (Appendix A). Extended clinical information is provided in Appendix A.

### 2.2. RNA-Seq Data Pre-Processing and Centroid Assignation

Retrieved RNA-seq data from selected patients included 23,501 genes, 16,993 of which remained after processing and filtering. Subsequently, the 2000 most variable genes were selected for further analysis. Samples lacking TCGA subtype information were assigned to CIN, MSI, GS, or EBV subtypes using Pearson correlation, and centroids were generated for each subtype. Following the assignment of previously unclassified samples to one of the four TCGA molecular subtypes, the overall distribution of the 142 patients was as follows: CIN 77 (54.2%), MSI 27 (19.0%), GS 29 (20.5%), and EBV 9 (6.3%) (Appendix A).

### 2.3. Functional Characterization of the TCGA GA Cohort Treated with Adjuvant CT

A network using PGM incorporating the 2000 most variable genes was built. Its functional structure was explored and 12 functional nodes with overrepresented biological functions were defined (Figure 1, Appendix A).

Previously established TCGA GA molecular subtypes were compared according to the activities of functional nodes. MSI tumors exhibited higher activity in the transcription functional node. CIN tumors presented higher activities in keratinocytes, membrane transport, Wnt pathway, and transcription functional nodes, while GS tumors showed higher activity in lipid metabolism, cellular adhesion and proteolysis, cellular adhesion and differentiation, and immune system and inflammatory response functional nodes (Appendix A).

### 2.4. Biological Layer Analysis

To explore the presence of different layers of information among the transcriptomics data, sparse k-means/CC algorithm workflow was used [20,21]. Seven biological layers were defined. All of the layers divided patients into two groups except layer 7, which divided them into three groups. Each layer was associated with specific biological functions using a gene ontology analysis. The characteristics of each layer are provided in Table 1.

The distribution of TCGA subtypes among the newly defined biological layers was then explored. Layers 1, 2, 4, 5, and 7 classified patients into similar groups to those previously defined by the TCGA classification (Table 2). However, proteolysis and lipid metabolism layers showed no relation with CIN, MSI, and GS subtypes, dividing samples into groups different from TCGA subtypes. The evaluation of the relationship between the TCGA EBV subtype and biological layers is hindered by the limited number of patients. Nonetheless, all EBV samples were classified into lipid metabolism cluster 2 and cellular adhesion cluster 3.

### 2.5. Combined Molecular Layer

As layers 1, 2, 4, and 5 provide overlapping information (Appendix A) and show correlation with the previously defined TCGA subtypes, they were grouped into a unique classification called combined molecular layer (CML). Since layer 7 divided patients into similar groups and had similar biological functions to layer 2, its information was disregarded. A total of 456 genes from layers 1, 2, 4, and 5 were used to define the CML (Appendix A) and 3 groups were established using CC analysis: 45 (31.69%) samples were included in CML1, 39 (27.46%) in CML2, and 57 (40.14%) in CML3.

CML3 tumors showed the highest lipid metabolism, keratinocytes, and Wnt pathway functional node activities, whereas CML2 presented the highest cellular adhesion, immune system, and proteolysis functional node activities (Appendix A). Survival analysis showed significantly different outcomes among these groups: CML1 had a better prognosis than CML2 and CML3, with CML2 emerging as the worst prognostic group (*p* = 0.01) (Figure 2A). When the CML classification was compared with the TCGA subgroups, it was observed that CML1 correlated closely to MSI tumors, CML2 to GS tumors, and CML3 to CIN tumors (Figure 2B).

#### CIN Molecular Subtype

CML3 includes 63.6% of TCGA CIN tumors, while 24.7% of them were assigned to CML1 along with the majority of MSI tumors (Figure 2B), thus suggesting transcriptomic heterogeneity among this subtype. When we explore this heterogeneity, differences in functional node activities between these two CIN subgroups were found. CML3 CIN tumors (CIN-classical tumors from now on) showed significantly higher cellular adhesion and proteolysis, immune system and inflammatory response, cellular adhesion and differentiation, transcription, Wnt pathway, and transmembrane transport functional node activities than CML1 CIN tumors (CIN-MSI-like tumors from now on) (Appendix A). Moreover, CIN-classical tumors presented a tendency to a worse OS than CIN-MSI-like (HR= 4.33; 95%CI 0.87–21.47, *p* = 0.07) (Figure 3).

### 2.6. Proteolysis Layer

The classification based on the proteolysis layer did not correlate with the TCGA GC molecular subtypes. It classified patients into two categories, consisting of 79 (56%) and 63 (44%) patients (clusters 1 and 2, respectively) (Table 1). Significant variations in functional node activities related to lipid metabolism, membrane function, cellular adhesion, proteolysis, inflammatory responses I and II, and keratinocytes were identified between these two groups (Appendix A). However, no differences in OS were observed, either independently or in combination with the previously defined CML classification (Appendix A).

Proteolysis categorization was established based on genes such as claudin-18, mucins 5 and 6, and keratin 20, among others. Expression levels of claudin-18 were significantly different between proteolysis clusters 1 and 2, resulting in low-claudin-18 and high-claudin-18 expression groups (Figure 4A). Additionally, claudin-18 expression divides patients into low-claudin-18 and high-claudin-18 expressors within each of the TCGA subtypes, after being separated by the proteolysis layer within these groups (Figure 4B).

### 2.7. Lipid Metabolism Layer

The lipid metabolism layer stratified samples into two groups: 70 patients (49%) in cluster 1 and 72 patients (51%) in cluster 2. Differences were observed in lipid metabolism, transcription, membrane, keratinocytes, immune systems, and inflammatory response II (Appendix A). However, OS showed no significant distinction between the two clusters (Appendix A).

Upon incorporating information from the lipid metabolism layer into the CML, variations in OS were identified within the CML2 cluster (as previously mentioned, closely aligned with the TCGA GS subgroup). The median OS was 60.37 months for CML2 (GS)-lipid metabolism 2 and 15.57 months for CML2 (GS)-lipid metabolism 1 clusters, respectively (HR = 2.88; 95% CI, 1.14–12.66; *p* = 0.0368) (Figure 5).

## 3. Discussion

GA represents a complex and clinically diverse malignancy with a significant global health impact. Even in localized or locally advanced stages, despite recent advances in multimodal treatment, the prognosis remains poor, with cure rates less than 40%. We need better tools to more effectively identify which patients will benefit from treatment and discover new therapeutic targets, aiming to improve these poor outcomes.

Several clinicopathological factors have been studied to better identify patients who may benefit from CT or who are chemoresistant. These include the neutrophil-lymphocyte or lymphocyte-monocyte ratio [9], systemic immune-inflammation index [22], lymphovascular and perineural invasion [23], tumor differentiation [24], tumor regression rate, and lymph node involvement [25]. However, few are commonly used in treatment decisions. From a molecular perspective, promising studies focus on the prognostic value of MSI [25], peritumoral inflammatory infiltrate [10], immune checkpoint expression [10], and circulating tumor DNA (ctDNA) after surgery [26]. Still, no definitive biomarkers or molecular profiles have been established in clinical practice to guide the need for chemotherapy.

Despite the insights provided by the TCGA work [11], its utility in treatment decision-making for GA patients is not clear. Clinical trials are typically not designed for specific molecular subgroups, and few studies have explored the prognostic significance of this classification [12,15], and its role in the selection of patients who may benefit from available treatments. In addition, these classifications are far from perfect, as the populations included are not homogeneous. New strategies for the early diagnosis of GA and the identification of biomarkers that enable a more precise selection of patients who may benefit from existing treatments are essential. Emerging tools, such as micro-RNA detection, hold potential as early diagnostic methods [27] and may also serve as predictors of treatment resistance [28,29], offering valuable insights for guiding personalized therapeutic strategies. In the context of RNA-seq analysis, various studies have identified subsets of GA in different stages with characteristic expression profiles that correlate with certain histopathological variables, a higher risk of recurrence, or specific patterns of dissemination, constituting independent prognostic predictors for this neoplasm [14,18]. However, these tools have not yet been incorporated into clinical practice.

This study aims to deeply explore the molecular heterogeneity of GA and its implications for CT response and survival. Using a multi-layered functional analysis approach on CT-treated localized GA patients from the TCGA cohort, we identified distinct functional layers, molecular subtypes, and survival patterns associated with specific biological functions. These findings enhance our understanding of the biology of the disease and hold potential implications for personalized treatment strategies.

Probabilistic graphical models (PGMs) offer a valuable tool for modeling gene interactions, which facilitate genetic data analysis and disease prediction. PGMs have proven to be effective in identifying variations in biological processes across diverse tumor types [20,30,31]. Similarly, classification methods like sparse k-means [32] and consensus cluster (CC) [33] have shown efficacy in delineating tumor and immune subtypes in breast, bladder, or colon cancers [20,21,34].

A notable strength of our study lies in the selection of a more homogeneous population compared to the broader TCGA cohort. Specifically, we included only patients with localized tumors who received 5-fluorouracil and/or platinum-based adjuvant chemotherapy. In our cohort, the median OS was not reached during a median follow-up of 15 months. However, it is crucial to acknowledge the dynamic nature of the treatment landscape for locally advanced GA. Patients in the TCGA study likely underwent less effective chemotherapy regimens, with over 65% receiving monotherapy involving fluoropyrimidines or platinum. This stands in contrast to contemporary, more aggressive approaches such as the perioperative FLOT schedule [7].

While the original TCGA study did not report survival differences by molecular subtypes in his overall population, our analysis focused on localized GA patients treated with adjuvant CT revealed significant distinct outcomes among TCGA subtypes. GS tumors showed the worst prognosis, aligning with other studies based on ACRG [12] or TCGA [11] classifications. Historically, the Lauren diffuse histological subtype, linked to GS tumors, has been linked to poorer survival in both localized and advanced diseases [35,36,37]. In the TCGA, the diffuse histology is strongly associated with the molecular GS subtype, suggesting an unfavorable prognosis. Reports have also highlighted the inherent chemoresistance of these tumors [35,38], emphasizing the need to explore alternative therapeutic approaches in this population.

In our study, retrieved RNA-seq data from the selected 142 patients initially covered 23,501 genes. To manage data complexity, we prioritized the analysis of the 2000 most variable genes with the aim of capturing dynamic expression while minimizing noise. PGMs allowed us to discern the functional structure of these genes and to define 12 functional nodes harboring overrepresented biological functions, thus providing a comprehensive perspective of crucial processes in GA pathology and underlying molecular mechanisms. This approach, previously shown to be successful across different tumor types [20,21,30,34], enhances the interpretability of the results and facilitates a deeper understanding of gene interactions (Figure 1).

The key breakthrough of our analysis lies in the identification of seven distinct biological layers of information among the transcriptomic data (Table 2). These layers spanned a wide spectrum of biological activities, including muscular and nervous functions, cellular adhesion, proteolysis, cellular differentiation, immune system responses, lipid metabolism, and keratinocyte activities, which can assist in obtaining molecular data applicable to clinical practice. When we explored the distribution of TCGA subtypes among the defined biological layers, most of them grouped patients in a similar way to the TCGA classification. However, proteolysis and lipid metabolism layers classified patients independently of TCGA subtypes, thus potentially providing supplementary insights for a more precise classification of these tumors.

As described above, the CML, composed of 456 genes, emerged as a critical determinant of TCGA-defined molecular subtypes (Figure 2B) and is closely associated with prognosis (Figure 2A). CML1 predominantly includes MSI tumors, exhibiting the best survival, while CML2 correlates with GS tumors, representing the worst prognostic group. CML3 encompasses most CIN tumors, demonstrating intermediate outcomes. These clusters also showed differential expression of genes from various nodes (Appendix A). For instance, CML3 tumors exhibit the highest Wnt pathway functional node activity. Aberrant activation of the Wnt signaling cascade in primary tumors, particularly in GA, plays a pivotal role in cancer progression, promoting epithelial-mesenchymal transition (EMT), enhancing tumor cell migration and invasion, facilitating the escape from dormant states at metastatic sites, and modulating immune surveillance and evasion [39]. Consequently, dysregulation of the Wnt pathway may represent a novel therapeutic target, especially for CIN tumors. Investigational inhibitors targeting various points along the Wnt-APC-Beta-catenin pathway have already been developed [40,41] and hold promise as a therapeutic strategy against metastatic spread.

According to the CML classification, most of CIN tumors were designated as CML3, but approximately one-third were grouped into the CML1 cluster, along with the majority of MSI tumors. This observation indicates transcriptomic heterogeneity within the CIN category. Based on variations in functional node activities, we identified two distinct subsets of CIN tumors, termed CIN-classical and CIN-MSI-like tumors. The former subgroup displayed heightened transcription, cellular adhesion, immune system response, Wnt pathway, and transmembrane transport node activities, along with worse OS compared to CIN-MSI-like tumors. TP53 mutations, traditionally linked with CIN tumors, have been associated with diminished immune activities and resistance to immunotherapy [42]. Nevertheless, our findings suggest that the CIN-MSI-like subgroup may exhibit molecular behavior more similar to MSI tumors. Despite the absence of defined DNA repair protein deficiencies, they might respond more favorably to immunotherapy while displaying increased resistance to CT, rendering them less suitable candidates for adjuvant treatment.

When combining the lipid metabolism layer with the CML, it specifically segregated the CML2 (GS-like) cluster into two clearly differentiated prognostic groups (Figure 5). This suggests the existence of two distinct populations within the CML2 subtype with highly differential expression of molecules related to lipid metabolism. This finding is clinically relevant, given the potential for targeted therapeutic interventions based on the combination of molecular subtypes and lipid metabolism status. Abnormal lipid metabolism is identified as a critical feature in gastric tumor cells. Several studies indicate that altered expression of genes linked to fatty acid synthesis or oxidation is significantly associated with malignant phenotypes, promotes metastasis, and induces drug resistance in GA cells by reshaping the tumor microenvironment [43,44,45]. Accordingly, inhibitors of key lipid enzymes, such as carnitine palmitoyltransferase 1B (CPT1B) and CPT2 or those involved in fatty acid oxidation, are currently under preclinical and clinical investigations. They could potentially play a role in reversing CT resistance in gastrointestinal tumors [46,47]. This is particularly relevant in GS tumors, which have traditionally shown resistance to conventional treatments.

A noteworthy aspect of our study is the identification of two groups with significantly distinct claudin-18 expression levels within the proteolysis layer. Proteolysis cluster 2 displayed elevated claudin-18 activity, suggesting potential benefits from anti-claudin therapies. As a vital component of gastric epithelial tight junctions, claudin 18.2 (CLDN 18.2) plays a fundamental role in cell-cell adhesion and migration, selectively allowing the paracellular flux of ions and small molecules between cells [48]. In normal tissues, CLDN18.2 epitopes within tight junction complexes are typically inaccessible. However, in tumor tissues, these epitopes are exposed on the surface of tumor cells due to disrupted cell polarity during malignant transformation. CLD18-ARHGAP26/6 fusions, identified in GA, are predominantly present in the TCGA GS and diffuse-type subgroups [49], and closely correlated with high levels of CLDN 18.2 expression [50]. This expression is found in around 25–40% of metastatic gastroesophageal adenocarcinomas [51,52]. Recently, two phase 3 trials (GLOW [53] and SPOTHLIGHT [54]) revealed that zolbetuximab, a first-in-class monoclonal antibody binding to CLDN18.2, in combination with first-line CT, prolonged OS in HER2-negative and CLDN18.2-positive GA patients. Moreover, other treatment strategies targeting CLDN18.2 are under evaluation in both preclinical and clinical trials. Further studies are required to confirm whether these clusters with differential claudin-18 activity are better predictors of claudin expression than the TCGA subgroups and would allow us to better select patients who would have a greater benefit. This is crucial as CLDN18.2 positivity appears evenly distributed across the four molecular subtypes (Figure 4B). It also remains to be determined whether they play a predictive role in the response to anti-claudin therapies.

Our study has several limitations. The sample size is relatively small, and the limited follow-up may constrain the robustness of our conclusions regarding recurrence and survival outcomes. Moreover, variations in chemotherapy regimens received by patients could differ from current standard treatments, potentially influencing study outcomes. The inherent heterogeneity of GA presents challenges in precisely defining molecular subtypes and identifying specific biomarkers for treatment response. Additionally, the retrospective nature of our analysis limits our ability to establish causality and may introduce selection biases. The evolving landscape of treatment modalities in locally advanced GA adds complexity, as patients in our study may have received less active chemotherapy regimens compared to contemporary approaches. Nevertheless, our study provides valuable insights that enhance understanding of GA biology and may guide personalized treatment strategies.

Future molecular perspectives should include comprehensive genomic profiling to identify novel mutations and biomarkers that guide treatment decisions. Integrating multi-omics approaches can elucidate interactions between molecular pathways, particularly lipid metabolism and proteolysis. Additionally, focusing on the tumor immune microenvironment may help identify immune signatures predictive of immunotherapy responses. Exploring circulating biomarkers like micro-RNAs through liquid biopsies can enable real-time monitoring of treatment efficacy. Lastly, targeting cancer stem cells and investigating novel drug combinations could enhance therapeutic outcomes and address chemoresistance in GA.

## 4. Materials and Methods

### 4.1. Patient Selection and RNA-Seq Data Pre-Processing

We select patients with localized GA included in the TCGA project who received 5-fluorouracil and/or platinum-based adjuvant CT and have RNA-seq data available [9]. The patient selection process is illustrated in Appendix A.

RNA-seq values underwent log2 transformation, and genes with ≥75% of valid values were included in the analysis. Missing values were imputed using Perseus Software v12 [55], and the study focused on the top 2000 most variable genes based on standard deviation.

### 4.2. Molecular Subtype Attribution

A differential gene set among TCGA subtypes was defined using multi-class Significance Analysis of Microarrays (SAM). This involved samples with available TCGA molecular subtype information and the 2000 most variable genes. Centroids for each molecular subtype were then established. For tumors lacking TCGA molecular subtype data, gene expression values were compared with each subtype centroid using Pearson correlation, and the sample was assigned to the subtype with the highest correlation.

### 4.3. Network Construction and Functional Node Activities

A PGM using GA TCGA RNA-seq data was defined with the gRapHD R Package v6.0.1 [56]. The PGM aimed to minimize the Bayesian Information Criterion (BIC) in a high-dimensionality context. This was achieved through a two-step process: first, defining the minimum spanning tree with maximum likelihood, and then conducting a forward search of edges to minimize the BIC while preserving graph decomposability [57]. The acquired network was divided into branches, and functional nodes were determined based on each branch’s overrepresented function. Overrepresented functions were identified through gene ontology analyses using the DAVID webtool, specifying categories as GOTERM-DIRECT, Biocarta, and KEGG_PATHWAY, with “Homo sapiens” as the background [58]. Subsequently, the functional node activity for each node in each sample was calculated as the mean expression of genes associated with the primary function of that node. These functional node activities facilitated group-wise comparisons of samples.

### 4.4. Biological Layer Analysis

Sparse k-means [32] and Consensus Cluster (CC) algorithms [59] were employed for the molecular characterization of TCGA GA tumor samples [20,21]. Sparse K-means assigned weights to genes based on their relevance in sample classification, while CC determined the optimum number of sample groups defined by these genes. Genes identified in each layer underwent gene ontology analysis for functional information retrieval. Both analyses were performed recursively, defining different layers of information. Sparse K-means and CC calculations used R v3.2.5 with the sparcl package [32], and the Consensus Cluster Plus package v1.38 [59], respectively.

### 4.5. Differential Gene Expression Analyses

SAM was employed to analyze differential expression patterns among groups, with a false discovery rate (FDR) set below 5% [60]. Hierarchical clusters were constructed using an average linkage clustering method with Pearson correlation as distance. These analyses were conducted using the TM4 Multiexperiment Viewer (MeV) 4.9 software [61].

### 4.6. Statistical Analyses

Mann–Whitney, Kruskal–Wallis, and ANOVA’s multiple comparison tests were utilized to assess differences in functional node activity between groups, and results were presented using Tukey’s box and whiskers graphs. Differences in distributions were evaluated using the Chi-squared test, with a Yates correction or a Fisher exact test applied when necessary. Survival analyses employed Kaplan Meier and log-rank tests, defining OS as the time elapsed between diagnosis and death or last follow-up. GraphPad Prism v6 software was used for these analyses, and network visualization was performed using Cytoscape software v3.5.1 [62]. All *p*-values were two-sided and a *p*-value below 0.05 was considered statistically significant.

## 5. Conclusions

In summary, our comprehensive analysis of the TCGA GA cohort has revealed the presence of distinct functional layers, molecular subtypes, and survival patterns linked to specific biological functions. The identification of the CML and its interaction with lipid metabolism status provides novel perspectives into patient prognosis, particularly in the context of the GS molecular subtype. Additionally, the discovery of claudin-18 activity and its potential as a therapeutic target adds further depth to our understanding of the disease. All these findings pave the way for more precise and personalized treatment strategies and help clinicians and researchers develop targeted therapies that take into account the intricate biological complexity of this challenging malignancy. Further exploration of these layers and their associations with patient outcomes will undoubtedly shape the future of GA management, with the potential to improve survival rates and enhance the quality of life for affected individuals.

## Figures and Tables

**Figure 1 ijms-25-11371-f001:**
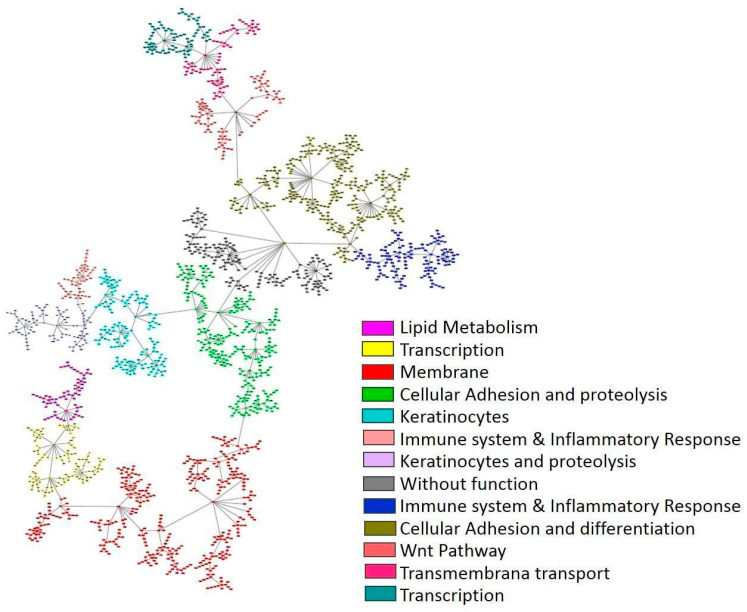
Network built using PGM and TCGA RNA-seq data from 142 localized gastric adenocarcinoma patients.

**Figure 2 ijms-25-11371-f002:**
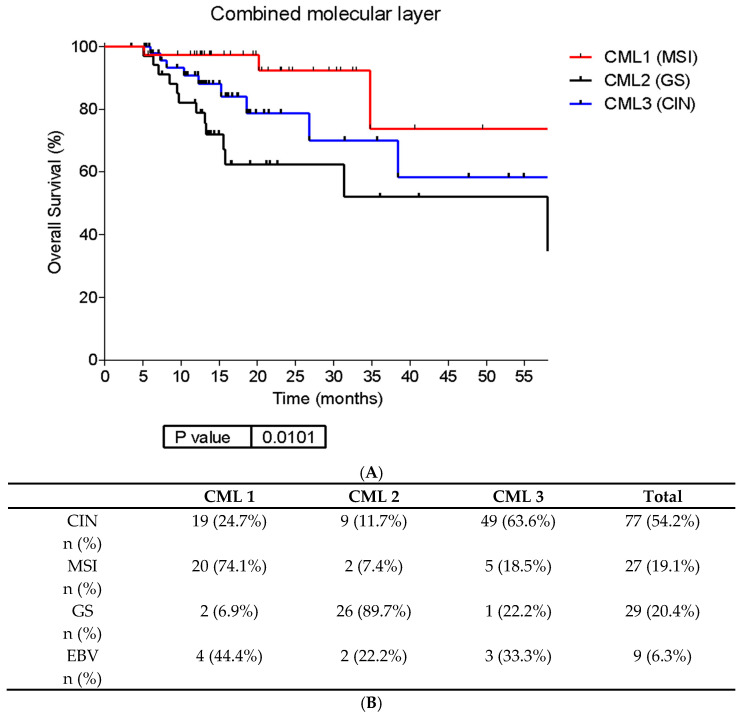
Molecular combined layer reflecting layers 1, 2, 4, and 5 information in TCGA GC cohort. (**A**) Survival analysis according to CML subgroups. (**B**) Distribution of the TCGA GC molecular subtypes in the CML groups. Abbreviations: CIN: chromosome instability. CML: combined molecular layer; EBV: Epstein–Barr virus; GS: genomically stable; MSI: Microsatellite-instability.

**Figure 3 ijms-25-11371-f003:**
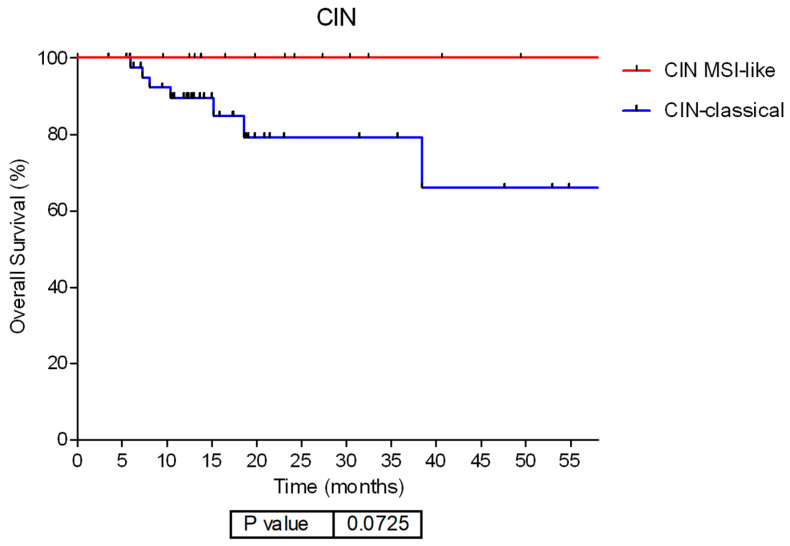
Molecular combined layer (CML) reflecting differences in functional node activity levels in the CIN TCGA subtype. Survival curves according to CIN subtype CML classification by functional information. Abbreviations: CIN: chromosome instability. MSI: Microsatellite-instability.

**Figure 4 ijms-25-11371-f004:**
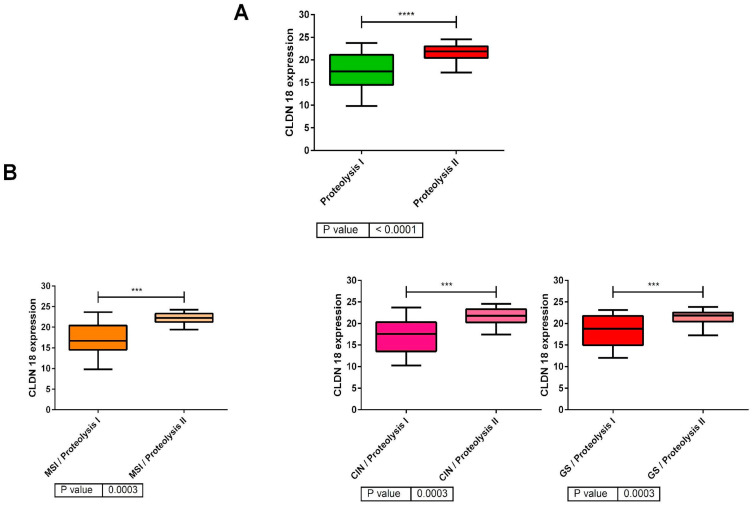
(**A**) Division of patients into low-claudin-18 and high-claudin-18 expression groups within the proteolysis layer (79 samples in Cluster 1 and 63 samples in Cluster 2). (**B**): Claudin-18 expression across the TCGA molecular subtypes (CIN, MSI, and GS) whose distribution according to the proteolysis layer is included in Table 2. ****: *p* < 0.0001; ***: *p* < 0.001.

**Figure 5 ijms-25-11371-f005:**
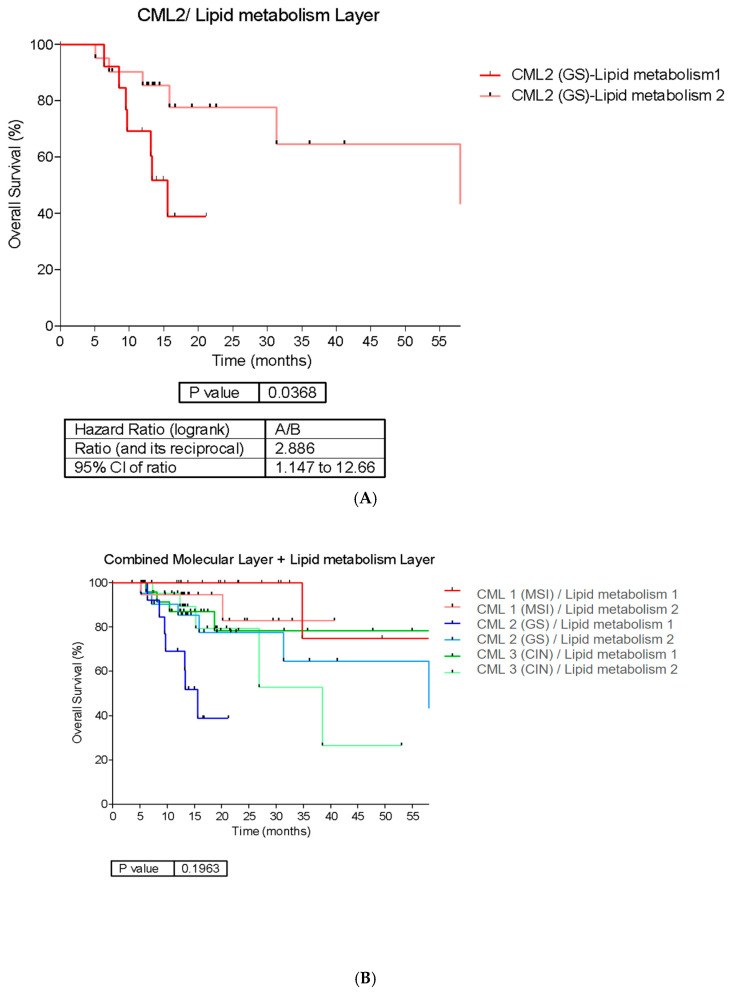
(**A**) Overall survival curves of patients classified into CML2 (aligned to GS tumors as shown in Figure 2B) with further stratification into Lipid Metabolism Cluster 1 and Lipid Metabolism Cluster 2. CML 2-Lipid Metabolism Cluster 2 exhibited higher median OS than Cluster 1. (**B**) Survival curves for the TCGA subtypes (MSI, GS, and CIN, corresponding to CML 1, CML 2, and CML 3, respectively, as shown in Figure 2B) stratified further by whether they belong to Cluster 1 or 2 of the lipid metabolism layer.

**Table 1 ijms-25-11371-t001:** Summary of biological layers characteristics.

Layer	Main Function (Gene Ontology)	Groups	Genes	Patients n (%)		
1	Muscular and nervous	2	118	Group 1 86 (60.6)	Group 2 56 (39.4)	
2	Cellular adhesion	2	97	Group 1 69 (48.6)	Group 2 73 (51.4)	
3	Proteolysis	2	46	Group 1 79 (55.6)	Group 2 63 (44.4)	
4	Cellular differentiation	2	125	Group 1 64 (45.1)	Group 2 78 (54.9)	
5	Immune system	2	116	Group 1 77 (54.2)	Group 2 65 (45.8)	
6	Lipid metabolism	2	65	Group 1 70 (49.3)	Group 2 72 (50.7)	
7	Cellular adhesion	3	104	Group 1 45 (31.7)	Group 2 52 (36.6)	Group 3 45 (31.7)

**Table 2 ijms-25-11371-t002:** Distribution of gastric adenocarcinoma TCGA molecular subtype in each layer.

Layer	Cluster	CIN n (%)	MSI n (%)	GS n (%)	EBV n (%)
Layer 1: Muscular and Nervous *	I	**54 (70.1)**	**23 (85.2)**	3 (10.3)	6 (66.7)
	II	23 (29.9)	4 (14.8)	**26 (89.7)**	3 (33.3)
Layer 2: Cellular Adhesion *	I	36 (46.8)	**23 (85.2)**	3 (10.3)	**7 (77.8)**
	II	41 (53.2)	4 (14.8)	**26 (89.7)**	2 (22.2)
Layer 3: Proteolysis	I	39 (50.6)	17 (63.0)	16 (55.2)	7 (77.8)
	II	38 (49.4)	10 (37.0)	13 (44.8)	2 (22.2)
Layer 4: Cellular Differentiation *	I	26 (33.8)	**24 (88.9)**	7 (24.1)	**7 (77.8)**
	II	**51 (66.2)**	3 (11.1)	**22 (75.9)**	2 (22.2)
Layer 5: Immune System *	I	**51 (66.2)**	**20 (74.1)**	1 (3.4)	5 (55.6)
	II	26 (33.8)	7 (25.9)	**28 (96.6)**	4 (44.4)
Layer 6: Lipid Metabolism	I	43 (55.8)	13 (48.1)	14 (48.3)	0 (0)
	II	34 (44.2)	14 (51.9)	15 (51.7)	**9 (100.0)**
Layer 7: Cellular Adhesion	I	44 (57.1)	1 (3.7)	0 (0)	0 (0)
	II	20 (26.0)	4 (14.8)	**28 (96.6)**	0 (0)
	III	13 (16.9)	**22 (81.5)**	1 (3.4)	**9 (100)**

Abbreviations: CIN: chromosome instability. EBV: Epstein–Barr virus; GS: genomically stable; MSI: Microsatellite-instability. * Layers included in the combined molecular layer. Statistically significant results are marked in bold.

## Data Availability

The RNA sequencing data analyzed in this study are available from The Cancer Genome Atlas (TCGA) database (https://cancergenome.nih.gov/, accessed on 1 June 2023) upon request. Access to the data is subject to approval from TCGA. Additional details regarding data availability and access can be obtained from the corresponding author.

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
