# Peer review of "Layer Analysis Based on RNA-Seq Reveals Molecular Complexity of Gastric Cancer"

_ijms, 2024, doi:10.3390/ijms252111371_

Round 1
Reviewer 1 Report
Comments and Suggestions for Authors
The article "Layer Analysis Based on RNA-Seq Reveals Gastric Cancer’s Molecular Complexity" brings new insights into gastric cancer and analyzes its molecular complexity. Recommendations:
1. The introduction contains many elements that should be moved to the discussion section.
2. The materials and methods section should not include citations. Additionally, add a flowchart and detail the inclusion and exclusion criteria.
3. The statistical analysis is good, but the figures contain a lot of information that repeats data from the statistics. Calculate the power of the study.
4. The discussions should be expanded and more recent articles should be added. Also, discuss future perspectives, especially considering that new micro-RNA markers are becoming increasingly useful: I recommend the article – 10.3390/ijms25147898. Moreover, besides molecular mechanisms, imaging techniques are advancing hand-in-hand, with a focus on EUS for differential diagnoses in the gastric area – I recommend the article - 10.3390/diagnostics14070675.
5. Add a conclusions section.
Author Response
The article "Layer Analysis Based on RNA-Seq Reveals Gastric Cancer’s Molecular Complexity" brings new insights into gastric cancer and analyzes its molecular complexity. Recommendations:
-Comments 1: The introduction contains many elements that should be moved to the discussion section.
-Response 1: We have moved several elements from the introduction to the discussion section, including information about probabilistic graphical models and classification methods, as they are more appropriate for that part of the manuscript. We have also retained a statement about the limitations of these classifications, highlighting the issues of population heterogeneity and unclear prognostic categorizations. This restructuring aims to enhance the clarity and focus of both sections.
-Comments 2: The materials and methods section should not include citations. Additionally, add a flowchart and detail the inclusion and exclusion criteria.
-Response 2: We have added a flowchart to explain the sample selection process (Supplementary Figure 1), which is now referenced in the text under the Materials and Methods section as, “The patient selection process is illustrated in Supplementary Figure 1.” Additionally, we have reduced the number of citations in the Materials and Methods section. However, we have retained citations related to the software and analyses employed, particularly those designed by third parties, as we believe these are essential for transparency. Our group has previously published in this journal (Int J Mol Sci. 2023 Jan 2;24(1):801. doi: 10.3390/ijms24010801), and in our prior paper similar citations were included in the Methods section. Nonetheless, we will fully adhere to the editor’s criteria on this matter.
- Comments 3: The statistical analysis is good, but the figures contain a lot of information that repeats data from the statistics. Calculate the power of the study.
- Response 3: This is an exploratory study aimed at molecularly characterizing gastric adenocarcinoma, rather than a clinical trial designed to test specific interventions or hypotheses. Therefore, the conventional concept of calculating study power, typically applied to hypothesis-driven studies with pre-specified endpoints, does not directly apply. Our goal was to generate novel biological insights and hypotheses, which can inform future research and clinical trials.
- Comments 4: The discussions should be expanded and more recent articles should be added. Also, discuss future perspectives, especially considering that new micro-RNA markers are becoming increasingly useful: I recommend the article – 10.3390/ijms25147898. Moreover, besides molecular mechanisms, imaging techniques are advancing hand-in-hand, with a focus on EUS for differential diagnoses in the gastric area – I recommend the article - 10.3390/diagnostics14070675.
- Response 4: Thank you for your valuable comments. We acknowledge the importance of expanding the discussion and incorporating more recent literature, particularly on emerging biomarkers techniques, like micro-RNA together with other multi-omics tools that may play a role in prognosis and patient selection for treatment. We will include the recommended references, particularly the study on micro-RNA markers (10.3390/ijms25147898), which highlights the growing relevance of micro-RNAs in gastric cancer prognosis and therapy. This addition will complement our molecular findings and provide insights into the evolving role of micro-RNAs as diagnostic and prognostic tools. This topic aligns with the future perspectives section, as these technologies will likely impact clinical decision-making and patient stratification in gastric adenocarcinoma. We have also expanded the discussion on future perspectives, focusing on emerging molecular markers, such as micro-RNAs, the relationship between lipid metabolism and gastric cancer biology, the potential of claudin-18 expression and other markers like immune microenvironments as biomarkers or targeting cancer stem cells and investigating novel drug combinations.
-Comments 5: Add a conclusions section.
-Response 5: We have added a conclusions section
Reviewer 2 Report
Comments and Suggestions for Authors
The study demonstrates that RNA-seq analysis revealed the gene expression diversity in gastric cancer.
1. The title should be revised to change "Rna-Seq" into "RNA-Seq" and "Gastric Cancer's Molecular Complexity" into "Molecular Complexity of Gastric Cancer" etc.
2. Results need to be revised:
Section 3.2. RNA-Seq data pre-processing and centroid assignation can be expanded. The number and detailed information of CIN, GS, EBV and MSI samples may be shown in a Table.
3. Figure 4 needs some more explanation and higher resolution. Current graphs do not seem to have significant differences. The number of samples and detailed sample information are needed.
4. Figure 5 needs to be revised to explain what "combined" means in the graph. CML2 is shown as CML II in Figure 2B. Please make it consistent. The differences between "Cluster" in Figure 2A and "Combined" in Figure 5 are unclear. More detailed descriptions are needed in figure legends.
Comments on the Quality of English Language
RNA sequencing is sometimes shown as RNA-seq or RNA sequencing. Please make it consistent in the manuscript.
Author Response
-Comments 1: The title should be revised to change "Rna-Seq" into "RNA-Seq" and "Gastric Cancer's Molecular Complexity" into "Molecular Complexity of Gastric Cancer" etc.
-Response 1: We have revised the title as suggested, changing "Rna-Seq" to "RNA-Seq" and "Gastric Cancer's Molecular Complexity" to "Molecular Complexity of Gastric Cancer."
-Comments 2. Results need to be revised: Section 3.2. RNA-Seq data pre-processing and centroid assignation can be expanded. The number and detailed information of CIN, GS, EBV and MSI samples may be shown in a Table.
-Response 2: Thank you for the valuable feedback. We have revised section 3.2 to provide a more detailed explanation of the RNA-seq data pre-processing and centroid assignment. We have already included the percentage of samples assigned to each group in Supplementary table 1. We have added this information as follows: “Following the assignment of previously unclassified samples to one of the four TCGA molecular subtypes, the overall distribution of the 142 patients was as follows: CIN 77 (54.2%), MSI 27 (19.0%), GS 29 (20.5%), and EBV 9 (6.3%). This is detailed in Supplementary Table 1”.
-Comments 3: Figure 4 needs some more explanation and higher resolution. Current graphs do not seem to have significant differences. The number of samples and detailed sample information are needed.
-Response 3: We acknowledge that Figure 4 requires a clearer presentation of the differences in claudin-18 expression. We have updated the figure to improve resolution and ensure that the graphs clearly display the distinct expression patterns between the two groups defined by the proteolysis layer, as well as within each TCGA subtype.
The number of samples belonging to each layer is already expressed in the original manuscript in section 3.6: “The classification based on the proteolysis layer did not correlate with the TCGA GC molecular subtypes. It classified patients into two categories, consisting of 79 (56%) and 63 (44%) patients (cluster 1 and 2, respectively)”.
Similarly, we have revised the legend of Figure 4 to enhance its clarity, as follows: “Figure 4A: Division of patients into low-claudin-18 and high-claudin-18 expression groups within the proteolysis layer (79 samples in Cluster 1 and 63 samples in Cluster 2). Figure 4B: Claudin-18 expression across the TCGA molecular subtypes (CIN, MSI, GS, and EBV), whose distribution according to the proteolysis layer is included in Table 2”.
We believe these improvements will highlight the differences in claudin-18 expression more effectively and address the concerns about its significance.
-Comments 4: Figure 5 needs to be revised to explain what "combined" means in the graph. CML2 is shown as CML II in Figure 2B. Please make it consistent. The differences between "Cluster" in Figure 2A and "Combined" in Figure 5 are unclear. More detailed descriptions are needed in figure legends.
-Response 4: We have updated the term "combined" in the graph to reflect its meaning more clearly. "Combined" in Figure 5 refers to the integration of information from the lipid metabolism layer with the CML (Combined Molecular Layer) classification. Specifically, this figure illustrates survival outcomes after incorporating lipid metabolism data into the CML2 group (closely aligned with the GS subtype from the TCGA classification).
Additionally, we have standardized the nomenclature, changing CMLII in Figure 2A into CML 2 to match the labeling used in Figure 2B for consistency.
The differences between "Cluster" in Figure 2A and "Combined" in Figure 5 are as follows:
- In Figure 2A, "Cluster" refers to the groups derived purely from the CML classification.
- In Figure 5, "Combined" refers to the clusters derived by integrating both CML classification and lipid metabolism layer data.
We have provided a more detailed description in the figure legends to ensure these distinctions are clearly communicated, as follows: “Figure 5. (A) Overall survival curves of patients classified into CML2 (aligned to GS tumors as shown in Figure 2B) with further stratification into Lipid Metabolism Cluster 1 and Lipid Metabolism Cluster 2. CML 2-Lipid Metabolism Cluster 1 exhibited a higher median OS than Cluster 2. (B) Survival curves for the TCGA subtypes (MSI, GS, and CIN, corresponding to CML I, CML II, and CML III, respectively, as shown in Figure 2B) stratified further by whether they belong to Cluster 1 or 2 of the lipid metabolism layer”
Reviewer 3 Report
Comments and Suggestions for Authors
The manuscript by Perez-Wert et al. offers a thorough analysis of the TCGA GA cohort, revealing distinct functional layers, molecular subtypes, and survival trends associated with specific biological processes. The discovery of the CML as well as its relationship with lipid metabolism status offers new insights into patient prognosis, especially regarding the GS molecular subtype. Furthermore, the identification of claudin-18 activity and its possible role as a therapeutic target enhances our comprehension of the disease.
The manuscript is nicely presented and well written. It adds some new layer of information.
I have a few suggestions for polishing the manuscript.
Comments:
11. The authors should simplify their result. They should explain it in a more concise manner.
22. Among 142 patients selected for analysis, what is the percentage of male and female patients?
3. The EBV subtype is limited to a small number of patients. Did the authors think they overestimated the classification based on lipid metabolism in EBV because of the limited number of patients?
4. Did the authors only look at the CML classification for gastric cancer? Did they perform their analysis on other types of cancers?
5. Did the authors find any unique biomarkers apart from Claudin 18?
Author Response
The manuscript by Perez-Wert et al. offers a thorough analysis of the TCGA GA cohort, revealing distinct functional layers, molecular subtypes, and survival trends associated with specific biological processes. The discovery of the CML as well as its relationship with lipid metabolism status offers new insights into patient prognosis, especially regarding the GS molecular subtype. Furthermore, the identification of claudin-18 activity and its possible role as a therapeutic target enhances our comprehension of the disease.
The manuscript is nicely presented and well written. It adds some new layer of information.
I have a few suggestions for polishing the manuscript.
Comments:
- Comments 1:. The authors should simplify their result. They should explain it in a more concise manner.
- Response 1: We’ve made our best effort to be more concise in the results presentation.
- Comments 2: Among 142 patients selected for analysis, what is the percentage of male and female patients?
- Response 2: 95 (66.9%) patients are male and 47 (33.1%) are female. This data is included in the Supplementary Table 1 (summary of clinical characteristics of the cohort). This table has to be included as supplementary material due to the space limitations.
- Comments 3: The EBV subtype is limited to a small number of patients. Did the authors think they overestimated the classification based on lipid metabolism in EBV because of the limited number of patients?
- Response 3: Unfortunately, as the reviewer points out, the number of EBV patients is very limited, thus it is not possible to establish any final conclusion about the behavior of this group.
- Comments 4: Did the authors only look at the CML classification for gastric cancer? Did they perform their analysis on other types of cancers?
- Response 4:
We have used this layer analysis strategy to analyze other cancer subtypes with interesting results. For instance, in muscle-invasive bladder carcinoma, we established that it is possible to independently classify tumors according to immune features or luminal/basal characteristics, opening the possibility to identify patients that may be candidates to immunotherapy more accurately (Trilla-Fuertes et al. BMC Cancer 2019). Another example is melanoma, where we established that this layer analysis identified a group of patients who are going to respond to anti-PD1 inhibitors (Trilla-Fuertes et al. Int J Mol Sci 2023). In summary, this layer method classification allows us to perform a deeper molecular characterization of tumors that is useful to predict response to treatment or propose new targeted therapies.
- Comments 5: Did the authors find any unique biomarkers apart from Claudin 18?
- Response 5: We identified a list of genes characteristic of each classification. However, to our knowledge, none of these genes are clearly actionable as occurs with claudin-18. However, we have included the lists of genes in which each classification is based on as a supplementary table, to make possible for other researchers to consult them.
Round 2
Reviewer 2 Report
Comments and Suggestions for Authors
The manuscript has been revised according to the reviewer's comments.
Very minor editing is needed:
1. In Figure 4B, the graphs show the claudin-18 expression in MSI, CIN, and GS, while the legend describes the TCGA molecular subtypes CIN, MSI, GS, and EBV. Please manage the description on EBV.
2. In Figure 5A, the explanation for labels for GS/6.1 and GS/6.2 is needed. Current legends indicate that CML2-Lipid Metabolism Cluster I exhibited a higher median OS than Cluster II. At the same time, the median OS was 60.37 months for CML2 (GS)-lipid metabolism 1 and 15.57 months for CML2 (GS)-lipid metabolism 2 cluster, respectively, in the text.
Please carefully make revisions.
Author Response
-Comments 1: In Figure 4B, the graphs show the claudin-18 expression in MSI, CIN, and GS, while the legend describes the TCGA molecular subtypes CIN, MSI, GS, and EBV. Please manage the description on EBV.
- Response 1: Thank you for your valuable comment. We did not include the EBV group in Figure 4B due to the small number of available samples in this category. We acknowledge this error in the figure legend and have corrected it accordingly to accurately reflect the data presented.
- Comments 2: In Figure 5A, the explanation for labels for GS/6.1 and GS/6.2 is needed. Current legends indicate that CML2-Lipid Metabolism Cluster I exhibited a higher median OS than Cluster II. At the same time, the median OS was 60.37 months for CML2 (GS)-lipid metabolism 1 and 15.57 months for CML2 (GS)-lipid metabolism 2 cluster, respectively, in the text.
-Response 2: Thank you for bringing this to our attention. We apologize for the error in the text. The correct description is as follows: the median OS was 60.37 months for the CML2 (GS)-lipid metabolism 2 cluster and 15.57 months for the CML2 (GS)-lipid metabolism 1 cluster. We have revised the figure, updated the text, and modified the legend to ensure that this information is now clearly and accurately presented. The correct description is as follows: the median OS was 60.37 months for the CML2 (GS)-lipid metabolism 2 cluster and 15.57 months for the CML2 (GS)-lipid metabolism 1 cluster.
Reviewer 3 Report
Comments and Suggestions for Authors
The manuscript by Perez-Wert et al. offers a thorough analysis of the TCGA GA cohort, revealing distinct functional layers, molecular subtypes, and survival trends associated with specific biological processes. The discovery of the CML as well as its relationship with lipid metabolism status offers new insights into patient prognosis, especially regarding the GS molecular subtype. Furthermore, the identification of Claudin-18 activity and its possible role as a therapeutic target enhances our comprehension of the disease.
The authors have addressed all the previous comments. Thus, the manuscript can be accepted in its present form.
Author Response
-Comments 1: The manuscript by Perez-Wert et al. offers a thorough analysis of the TCGA GA cohort, revealing distinct functional layers, molecular subtypes, and survival trends associated with specific biological processes. The discovery of the CML as well as its relationship with lipid metabolism status offers new insights into patient prognosis, especially regarding the GS molecular subtype. Furthermore, the identification of Claudin-18 activity and its possible role as a therapeutic target enhances our comprehension of the disease.
The authors have addressed all the previous comments. Thus, the manuscript can be accepted in its present form.
-Response 1: Thank you very much for your positive feedback and for the thorough review of our manuscript. We are grateful for the constructive comments provided throughout the review process, as they have helped us improve the quality of our work. We are pleased to hear that the manuscript has met your expectations and is now considered suitable for acceptance in its present form.